# Effects of Material Nonlinearities on Design of Composite Constructions—Elasto-Plastic Behaviour

**DOI:** 10.3390/ma13071792

**Published:** 2020-04-10

**Authors:** Aleksander Muc

**Affiliations:** Institute of Machine Design, Cracow University of Technology, 31-155 Cracow, Poland; olekmuc@mech.pk.edu.pl

**Keywords:** physicalnon-linearities, elasto-plastic behaviour, experimental analysis, finite element analysis, buckling, composite pressure vessels, stress distributions in nozzles and reinforcement, optimisation

## Abstract

Usually, the design of composite structures is limited to the linear elastic analysis only. The experimental results discussed in the paper demonstrate the physical non-linear behaviour both for unidirectional and woven roving composites. It is mainly connected with the micromechanical damages in composite structures, particularly with the effects of matrix cracking modeled in the form of elastic-plastic physical relations. In the present paper, the effects of both physical and geometrical non-linearities are taken into account. Their influence on the limit states (understood in the sense of buckling or failure/damage) of composite structures is discussed. The numerical examples deal with the behaviour of composite pressure vessels components, such as a cylindrical shell and the reinforcement of the junction of shells. The optimisation method of the reinforcement thickness is also formulated and solved herein.

## 1. Introduction

It is well known that the polymeric matrix of fibre-reinforced plastics (FRP) exhibits significant non-linear stress/strain behavior (the σ–ε diagram). In many cases, the non-linearity may even be detected upon initial loading of the material and continue until catastrophic failure. This non-linear mechanical response is mostly due to the non-linear constitutive behavior of the polymer matrix, microcracking of the matrix material, fibre/matrix interface debonding, and interlaminar delamination. This effect may be additionally enhanced by fibre orientations especially for fibres oriented at 45°where a typical plastic hardening is observed. FRP composites have been reported to exhibit non-linear behavior in the transverse direction and especially in the shear direction (Hahn [1], Hahn and Tsai [2]; Koerber et al. [3]).

There has been a considerable amount of research performed on the non-linear behavior of woven fabric composites at the meso-mechanical level using numerical [4,5] and analytical approaches [6,7]. These models are concerned with the effects of individual damage mechanisms, such as micro-cracking and non-linear behavior of individual constituents, on the total nonlinear response of the composite.

Now, for FRPs, plasticity theory and Continuum Damage Mechanics (CDM) become more traditional approaches to establish the constitutive models [8,9].

A great number of elasto-plastic constitutive models consisting of different yielding functions, flow rules, and the variety of possible numerical implementation algorithms is discussed in Ref [10].

However, in the majority of research works as well as in design codes, it is a common practice to employ the geometrically and physically linear plate/shell theory in the analysis of composite structures. It may lead to the incorrect evaluation of a failure (damage) index, particularly for structural elements where a stress concentration occurs, e.g., holes, junctions of plates, shells, and their reinforcement. In addition, in the case of reinforcements as a delamination failure becomes a dominant failure mode for composite structures, a possible stress relaxation due to the physical non-linearity may change design results completely.

In the present paper, we extend previous studies and examine the effects of physical and geometrical non-linearities on the failure modes of composite structures focusing our attention on the following problems:1description of physical non-linearities arising due to micromechanical damage of composite structures via elasto-plastic relations,2their comparison with experimental results and the proposals of approximations of the experimental stress-strain relations,3buckling analysis of compressed cylindrical shells taking into account large deflection, first-order transverse shear deformation theory, and physical non-linearities,4stress concentration and *First-Ply-Failure* (FPF) studies at the junction of two cylindrical shells to model and analyse the stress concentration effects caused by nozzles placed in the cylindrical part of composite pressure vessels,5optimal design of the junction reinforcement thickness in order to reduce and equalise stress concentration factors around the nozzles, i.e., in the perpendicular and transverse directions.

In all presented above problems, it is assumed that the physical non-linearity is a result of fibre orientations and according to experimental investigations, the highest effects of non-linearities is observed for fibres oriented at 45°. The numerical studies are conducted with the use of the FE package where a laminate is built of a finite set of 2D FE in the thickness (z) direction. Each 2-D FE corresponds to the assumed fibre orientations—groups of layers having the identical fibre orientations are described with the use of one 2D FE along the thickness direction. The physical, non-linear law (in the form of plastic deformations) is created independently for each individual 2D FE corresponding to the prescribed fibre orientation. In each FE, the initiation of non-linear deformations is described with the use of one parametrical plastic flow potential in the form proposed by Sun Chen [11]. The form of physical relations used in the FE modelling of non-linear (plastic) behaviour of individual plies in the laminate is broadly discussed by Chen Sun [12].

## 2. Experimental Results—Approximations of Non-Linearities

Figure 1 shows the experimental axial stress/strain response of the tensile test specimens, i.e., the mean values of at least five tests for each type of composite materials. The stress-strain relations and elastic moduli were evaluated in the standard range specifiedin Polish and ASTM Standards—see Figure 2.

Let us note that the different notations of the axes were used in Figure 1. The relations between them can be easily obtained from the following equations:Tensile stress = Tensile force/(Thickness × Width),
Tensile strain = Elongation/Length.
and using dimensions of the specimens (i.e., thickness, width, and length) shown in Figure 2. However, it should be emphasised that for our purposes, the character of deformations (linearity versus nonlinearity) is the most important and significant effect and not particularly the numbers, since they are directly related to the type of composite materials.

The angles 0° and 45° denote the direction of pure tension along the axis parallel to the fibres and at the direction 45° to the fibres (shear), respectively. Let us note that even for fibres directed at 0°, the physical non-linearity can be observed—Figure 1b–d. For plain woven roving, the non-linear behaviour is mainly the result of the thickness of individual layers. For aramid fibres (Figure 1a), the thickness is equal to 0.08 mm, whereas for glass and carbon fibres (Figure 1b,c) the thickness is equal to 0.22 mm. It seems to be significant for the micromechanical damage of specimens. For specimens subjected to tension with fibres directed at 45°, the analysis of the stress–strain diagrams demonstrates always non-linearity, but this behaviour is not identical—compare Figure 1a,c,d with Figure 1b,e. Let us note that the non-linear deformations of specimens oriented at 45° depend also on the form of manufacturing—compare Figure 1a–d with Figure 1e (the pultrusion).

The approximations of the stress-strain diagram seem to be important in view of theoretical and numerical modelling, especially in the area of continuum damage mechanics. Two general approaches can be introduced:1the use of spline curves with a finite number of key-points—see Muc [16].2the definition of the σ–ε relation and fitting the parameters of this function using e.g., genetic algorithms—Muc [17].

Abramchuket al. [18] proposed the following form of the characteristics σ–ε:(1)$$\sigma =A(1-e^{-r\varepsilon })$$

For instance, after very rough approximations of the curves plotted in Figure 1d, one can find the following formulas: 0° orientation—σ = 2770(1 − exp(−0.159ε));
45° orientation—σ = 126(1 −exp(−2.113ε)). 

The distributions plotted in Figure 1b,e should be described with the use of more complicated functions than given in Equation (1).

## 3. Theoretical Description of Physical Non-Linearities

In general, in order to determine the yield and post-yield behavior of the material, any plasticity theory should cover the following fundamental points:1**Yield criteria:** to define specific stress combinations that will initiate the inelastic response, i.e., defining an initial yield surface.2**Hardening rule:** to define the evolution of the yield surface with stress, strain, and other parameters.3**Plastic stress-strain relation or a flow rule:** to relate the plastic strain increment to the current stress level and stress increment.

As it is demonstrated e.g., by Odegarda et al. [5], the definition of the yield surface is connected with the evaluation of lower number of experimental data than in the case of the hardening rule. Therefore, in the further discussion of non-linear problems (numerical), it is assumed that the hardening parameter is equal to zero. Thus, the non-linear problem considered herein can be called (in terms of theory of plasticity) as the elastic-perfectly plastic problem.

The character of the σ–ε plot imposes automatically that it is necessary to implement herein the plastic flow theory used in the physical description of isotropic materials. Sun Chen [11] proposed the use of one parametrical plastic flow potential (yield criterion) in the following form (the Kelvin–Voigt notation of tensors):(2)$$f=(\sigma_{2}^{2}+2a_{6}\sigma_{6}^{2})/2$$

Owen Li [19] have applied the plastic potential (yield criterion) similar to the Huber-Mises-Hencky yield condition, which has six parameters of anisotropy:(3)$$f=a_{1}{\left(\sigma_{1}-\sigma_{2}\right)}^{2}+a_{2}{\left(\sigma_{2}-\sigma_{3}\right)}^{2}+a_{3}{\left(\sigma_{3}-\sigma_{1}\right)}^{2}+3(a_{4}\sigma_{4}^{2}+a_{5}\sigma_{5}^{2}+a_{6}\sigma_{6}^{2})$$

The anisotropy parameters *a_1_ … a_6_* are taken from experimental data. Usually, the yield function *f* (Equations (2) and (3)) is equal to a material parameter *k*, i.e., *f(σ_ij_)* = k or it may be expressed in more general form *F(f,k)* = 0. It may be a constant (elastic perfect-plastic material) or can be defined as a function of a hardening parameter.

Let the total strain component be the sum of the elastic strain and the plastic one:(4)$$\left\{d\varepsilon \}=\{d\varepsilon^{el}\}+\{d\varepsilon^{p}\}=[S]\{d\sigma \}+\{d\varepsilon^{p}\right\}$$

The elastic strain increment is given from the Hook law, whereas the plastic one $\left\{d\varepsilon^{p}\right\}$ is derived from the plastic potential *f*—Equation (3):(5)$$\{d\varepsilon^{p}\}=d\lambda \left\{\frac{\partial f}{\partial \{\sigma \}}\right\}$$
and *d*λ is a proportionality constant. The symbol $\frac{\partial f}{\partial \{\sigma \}}$ can be represented in a vector form as follows: ${\left\{\frac{\partial f}{\partial \{\sigma \}}\right\}}^{T}=\left\{\frac{\partial f}{\partial \sigma_{1}},\frac{\partial f}{\partial \sigma_{2}},\frac{\partial f}{\partial \sigma_{3}},\frac{\partial f}{\partial \sigma_{4}},\frac{\partial f}{\partial \sigma_{5}},\frac{\partial f}{\partial \sigma_{6}}\right\}$. Differentiating the functional *F(σ_ij_, k)* with respect to two functions *σ_ij_* and *k*, the normality condition of the yield surface can be written as follows:(6)$$\frac{\partial F}{\partial \{\sigma \}}d\{\sigma \}-Ad\lambda =0,\;A=-\frac{1}{d\lambda }\frac{dF}{dk}dk,\frac{\partial F}{\partial \{\sigma \}}=\frac{\partial f}{\partial \{\sigma \}}$$

Multiplying both sides of Equation (4) by ${\left\{\frac{\partial f}{\partial \left\{\sigma \right\}}\right\}}^{T}[C]$, inserting Equation (6) and substituting the plastic deformations by Equation (5):(7)$${\left\{\frac{\partial f}{\partial \left\{\sigma \right\}}\right\}}^{T}[C]\{d\varepsilon \}={\left\{\frac{\partial f}{\partial \left\{\sigma \right\}}\right\}}^{T}[C][S]\{d\sigma \}+d\lambda {\left\{\frac{\partial f}{\partial \left\{\sigma \right\}}\right\}}^{T}[C]\left\{\frac{\partial f}{\partial \left\{\sigma \right\}}\right\}=d\lambda \left(A+{\left\{\frac{\partial f}{\partial \left\{\sigma \right\}}\right\}}^{T}[C]\left\{\frac{\partial f}{\partial \left\{\sigma \right\}}\right\}\right)$$

Using the above Equations (3), (5), and (7), one can obtain finally the incremental elastic-plastic flow rule that has the similar form as for isotropic materials, i.e.,
(8)$$\left\{d\sigma \}=[C](\{d\varepsilon \}-\{d\varepsilon^{p}\})=[C]\left(1-\frac{{\left\{\frac{\partial f}{\partial \left\{\sigma \right\}}\right\}}^{T}[C]}{A+{\left\{\frac{\partial f}{\partial \left\{\sigma \right\}}\right\}}^{T}[C]\left\{\frac{\partial f}{\partial \left\{\sigma \right\}}\right\}}\right)d\{\varepsilon \}=[C^{ep}]d\{\varepsilon \right\}$$
where [C] is the classical stiffness matrix and [C] = [S]^−1^.

## 4. Numerical Modelling

Designing with composite materials requires both designing of the structural construction and the material composition that makes the construction. The wall of composite pressure vessels is usually made of a polyvinyl chloride (PVC) liner (a corrosion layer) and structural layers made of a mat layer, woven roving layers, and filament winding layers—see Figure 3.

According to the classical shell theory of composite structures, an unsymmetric composition of wall thickness leads to the existence of bending even in the case of the internal pressure loading. In addition, the stress concentrators (boundary conditions: holes, junctions, variable wall thickness, etc.) may increase (or decrease) locally the flexural effects, which involves the necessity of numerical analysis. Therefore, the influence of fibre orientations on the structural behaviour of composite curved constructions cannot be directly transferred from the well known results valid for the membrane state only (the studies conducted by Muc [20]), especially in view of the failure analysis. The correct results can be obtained by numerical investigations only.

In the present work, the numerical analysis is conducted with the use of the numerical finite element (FE) package NISA II v. 17 [21]. In the analysis, nine noded quadrilateral FE denoted as NKTP 32 are implemented. The finite element NKTP 32 includes transverse shear effects of the first order (*G_13_* = *G_23_* = 0.5*G_12_*).

In general, the following assumptions are applied in the numerical model:1the normal shell displacements *w* are constant across the shell cross-section,2the normal stress components *σ_3_* (in the local-layer and global-laminate system of coordinates) are equal to zero,3the tensor [C] has five independent components,4the yield criterion Equation (3) is characterised by five anisotropic parameters, i.e.,
(9)$$f=a_{1}\sigma_{1}^{2}+2a_{12}\sigma_{1}\sigma_{2}+a_{2}\sigma_{2}^{2}+a_{3}\sigma_{4}^{2}+a_{4}\sigma_{5}^{2}+a_{5}\sigma_{6}^{2}$$

5both the stiffness matrix components [C] and the anisotropic parameters are subjected to the classical rule of transformation of the stress tensor (the rotation),6elastic piecewise linear hardening model can be applied in the analysis to define the more complicated description of the stress–strain curve (see e.g., Figure 1b)

Usually, physical non-linearities are connected with the gradual matrix cracking of composite constructions, which starts almost at the beginning of the loading process. Therefore, it is assumed that physical non-linearities occurs for loads equal to 0.2 P_max_, where P_max_ denotes the maximal value of loading corresponding to the analysedfibre orientations, i.e., 0°, 45°, etc.—see Figure 1.

The numerical results deal with two group of problems:1buckling of the compressed cylindrical shell with a rigid diaphragm—parametric studies limited to the analysis of structural composite layers only; see Figure 3, and2deformation analysis of pressure vessel components taking into account the existence of the liner; see Figure 3.

## 5. Non-Linear Buckling Analysis

Designing with composite materials involves the analysis of different modes of failure. Now, considering the buckling mode of failure of compressed laminated cylindrical shells, the large deformation analysis is carried out (geometrical non-linearities) including also physical non-linearities in the form discussed above. Rigorous analysis for the determination of stability conditions for anisotropic cylinders is discussed by Muc et al. [22].

The present results (Figure 4) exhibit the reduction of buckling loads comparing with the linear elastic approach only. Those effects are strongly dependent on fibre orientations in individual plies. The analysis is conducted for compressed cylindrical shells made of unidirectional FRP having the following stacking sequence [0_6_ ± 45_3_]_S_.

Figure 4 summarises the results of various numerical studies. All results are compared to the value of buckling loads obtained for the classical Love-Kirchhoff relations in order to observe the influence of kinematical assumptions on results. In general, numerical studies have been conducted for three different cases:1classical elastic shell analysis using 2D first-order transverse shear theory relations (FSDT),2geometrically non-linear elastic shell analysis using 2D first-order transverse shear theory relations (FSDT),3physically non-linear analysis shell analysis using 2-D first-order transverse shear theory relations; however, the physical non-linearities in the form given by Equations (2), (4) and (7) independently for each of individual plies in the laminate if the plasticity condition in Equation (2) is fulfilled.

The plots presented in Figure 4 demonstrate directly the influence of material properties (understood in the sense of the degree of orthotropy E_1_/E_2_) on the values of the buckling loads. As it is obvious, the Love-Kirchhof shell theory gives the wrong approximation of buckling loads. The geometrical non-linearities reduce the values of buckling loads comparing with the linear elastic FSDT buckling analysis; however, the difference is strongly dependent on the mechanical properties. As it is expected, physical non-linearities play the most important rule for plies oriented at 45°. The plastic effects reduce again buckling loads obtained for the linearised FSDT. Thus, the both effects of geometrical and physical non-linearities results in the reduction of buckling loads. However, the results are strongly dependent on ply orientations and the geometrical properties of shells, since the above-mentioned factors have an effect on the buckling mode.

## 6. Nozzle Openings

Now, let us consider the optimal design of reinforcements of the junction of two cylindrical shells—the nozzle-opening problem. Nozzle openings and reinforcements can be of two forms: the flush nozzle installation and the penetrating nozzle installation. The nozzle is usually attached to the main shell by hand lay-up reinforcements. It is assumed that for both cylindrical shells,the total shell thickness *t* is equal to 9.5 mm, and their outside radii *R* = 305 mm and *r* = 160 mm, respectively.

### 6.1. Stress Analysis of Nozzle Openings without Reinforcements

The finite element (FE) model and the geometry of the junction is presented in Figure 5. Two cross-sections are introduced, i.e., transverse and longitudinal, in order to present numerical FE results. The cylindrical shells are made of plain woven-roving plies (Figure 1c) constituting the structural layers (Figure 3) and having the following stacking sequence [0_6_ ± 45_3_]_S_.

In the figures, the following notation is used: outside—z = t/2, inside—z = −t/2 (Figure 3), x denotes the length measured along cross-sections (Figure 5) where x = 0 corresponds to the position of the junction (the negative values are measured along the nozzle) and s is the value of the stress related to the value of the internal pressure p.The dimensionless stress distributions are demonstrated in Figure 6 and Figure 7 and they show evidently the necessity of the junction shape optimisation understood in the sense of adding an additional material called the reinforcement.

Figure 6a and Figure 7a present the stress distributions taking into account elastic deformations only, whereas Figure 6b and Figure 7b exhibit the influence of plastic (non-linear deformations) of plies. As it may be seen, physical non-linearities result in the drastic reduction of the maximal stresses at the junction.

### 6.2. Design of Reinforcement

Shape optimisation of the shell reinforcement is connected with searching for the maximal critical loads that can be carried out by laminated structures, or it is introduced in order to equalise the stress (strain) failure criteria (in the sense of required yield condition or FPF criteria) around the boundaries or for the whole axisymmetric structure. However, it is not known in advance what type of failure criteria allows us to predict the lowest value of the failure load—see the discussion presented by Muc [23]. In addition, if we intend to equalise the values of the objective functional *U* (stress/strain criterion) along the junction C, then the optimisation problem differs qualitatively (the *MinMax* problem) from the typical formulations, leading to the limitation of the functional value, i.e., *Max U(x) ≤ U_admi_*_s_, where *U_admis_* denotes the admissible, prescribed a priori value of the functional. For the *MinMax* problems using the finite element method of solution, it is almost impossible to satisfy such a condition in an exact manner, since it is very difficult to obtain the identical value of the functional *U* at each nodal point. Figure 8 demonstrates the possible variations of the objective functional *U* for the initial and the final (optimal) shapes (see Muc [23]). Therefore, it is assumed that the solution is optimal as the following condition is satisfied:(10)$$U_{l}\leq U_{k}(s)\leq U_{u},U_{k}(x)=\sigma_{1}\varepsilon_{1}+\sigma_{2}\varepsilon_{2}+\sigma_{4}\varepsilon_{4}+\sigma_{5}\varepsilon_{5}+\sigma_{6}\varepsilon_{6},\;k=1,2,\ldots ,K$$
where: (11)$$U_{l}=(1-\alpha )U_{aver},\;\;\;\;\;U_{u}=(1+\alpha )U_{aver},\;\;\;\;\;U_{aver}=\frac{1}{K}\sum_{k=1}^{K}U_{k}(x)$$


α is a parameter defining the range of acceptance of the optimal solutions. In general, it is assumed that its value should be less or equal to 0.01 or 0.05. *U_k_(x)* denotes the value of the objective functional (the strain energy herein—Equation (10)) evaluated at the nodal point *k* along the junction. Thus, in the shape optimisation problems, the optimal values are reached with the prescribed values of acceptance α varying from 1% to 5%.

In the material optimisation problems, we are looking for the minimal weight (volume) of the structure, assuming in advance that the optimal structure satisfies all constraint conditions in the form of required failure, stress, strain, or deformation criteria.

The optimisation problem has been solved for the functional U in the form of the dimensionless stresses measured (similarly as previously) along the dimensionless distance x/R. x equal to zero corresponds to the junction, and the sign + describes the distance measured along the cylinder having the greater radius R, and the sign-describes the cylinder with the radius r. In fact, two independent optimisation problems have been solved: the first corresponds to the elastic deformations analysis only, whereas the second takes into account physical non-linearities in the form discussed in Section 3. Optimal design of the thickness reinforcement at the junction of two cylindrical shells (one of them represents a nozzle in a pressure vessel body) is conducted with the use of the Bezier spline functions. A broad discussion of the optimisation method as well as of the used genetic optimisation algorithms is presented by Muc and Gurba [17]. The objective of the optimisation is the following: to equalise the stress concentration factors around the nozzles including elastic and elastic-plastic deformation effects—it is formulated as the MinMax problem. It can be achieved by the growth of the reinforcement thickness ∆*t*—Figure 9. It is assumed that the thickness of the reinforcement is constant along the length *x*.

The results of the optimisation are plotted in Figure 10 and Figure 11, independently in the transverse and longitudinal directions since the stress distributions are completely different. The vertical axes characterise the relative changes of the reinforcement thicknesses, i.e., (t + ∆t)/t. The above-mentioned figures exhibit also differences between optimal reinforcements in the linear (elastic) and non-linear (elastic-plastic) cases.

As it may be seen, physical non-linearities change completely the optimal reinforcement distributions, since the stress distributions for elastic and elastic-plastic cases are completely different. However, on the other hand, the results obtained with the use of non-linear, elastic-plastic analysis demonstrate evidently the necessity of taking physical non-linearities into account in the design procedures of composite pressure vessels.

## 7. Conclusions

In the experimental analysis, for various composite materials, the stress-strain diagrams exhibit physical non-linearities. These effects arise due to the micromechanical damage of composite constructions, particularly matrix cracking. The necessity of taking into account of both geometrical and physical non-linearities in the design of pressure vessels made of fibre-reinforced plastics is pointed out in the present paper. Studying the presented above approaches, methods, and results, the following conclusions can be drawn:1The physical non-linearities can be described by elasto-plastic analysis.2The elasto-plastic analysis can be conducted with the use of the commercial FE packages, such as NISA II, Ansys, and Abaqus.3In the design of pressure vessels and pipings made of fibre-reinforced plastics, it is necessary to conduct FE analysis and in addition to take into account the directional dependence of the material properties.4The linear predictions of buckling pressures for pressure vessel components available in the design codes do not reflect satisfactorily the directional dependence of buckling pressures, since geometrical and physical non-linearities should be considered.5Taking into account non-linear physical relations, one may reduce significantly stress concentrations arising at the junction of cylindrical shells.6The reinforcement of the junction of cylinders may be implemented into optimisation analysis, and the thickness of the reinforcement may be better predicted.7In view of optimisation analysis, a multicriterion objective functional should be formulated in order to take into account various failure criteria, e.g., delaminations.

In summary, the present study shows evidently that the use of present design codes result in the wrong prediction of the pressure vessel components thicknesses due to the incorrect application of experimental data, demonstrating the existence of both physical and geometrical non-linearities. In our opinion, the proper conjunction of numerical analysis with the experimental data may result in the further weight savings of pressure vessels and in this way a better optimisation of such structures.

At the end of these considerations, it should be emphasised that various variants of the non-linear degradation of composite material properties exists by the visco-elastic modelling Muc, Krawiec [24] or by the progressive failure modelling implemented into FE packages (e.g., Ansys) Stawiarski [25].

## Figures and Tables

**Figure 1 materials-13-01792-f001:**
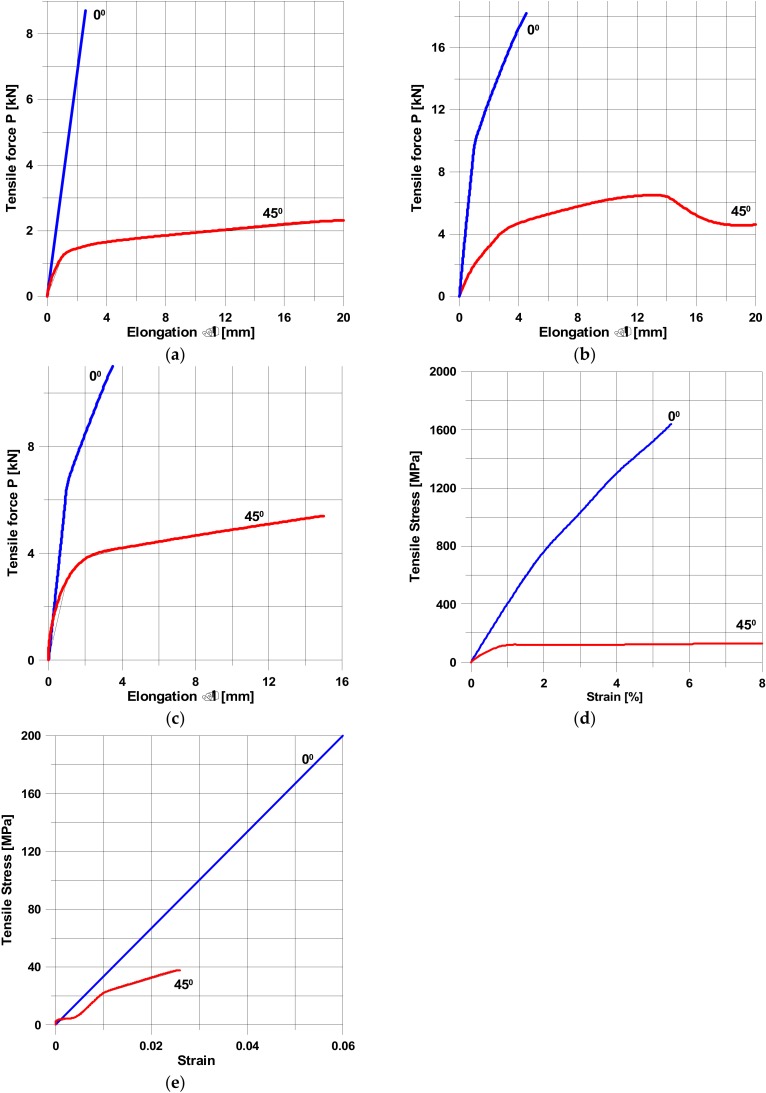
Stress/strain diagram of composites showing experimental responses. (**a**) plain woven roving—aramid/epoxy resin (Muc [13]). (**b**) plain woven roving—carbon/epoxy resin (Muc [13]). (**c**) plain woven roving—glass/epoxy resin (Muc [13]). (**c**) plain woven roving—glass/epoxy resin (Muc [13]). (**d**) unidirectional—glass/epoxy resin (Muc et al. [14]). (**e**) unidirectional—glass/vinyl ester resin (the pultrusion method—Muc et al. [15]).

**Figure 2 materials-13-01792-f002:**
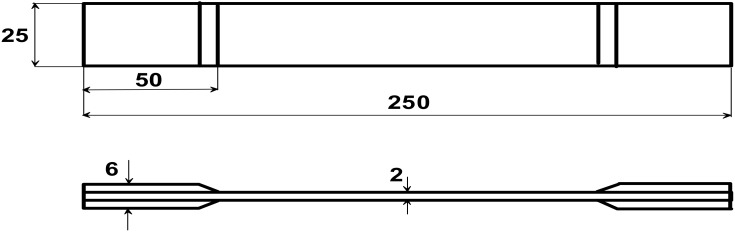
The form of the specimens used in tensile tests PN-EN-2561-1999.

**Figure 3 materials-13-01792-f003:**
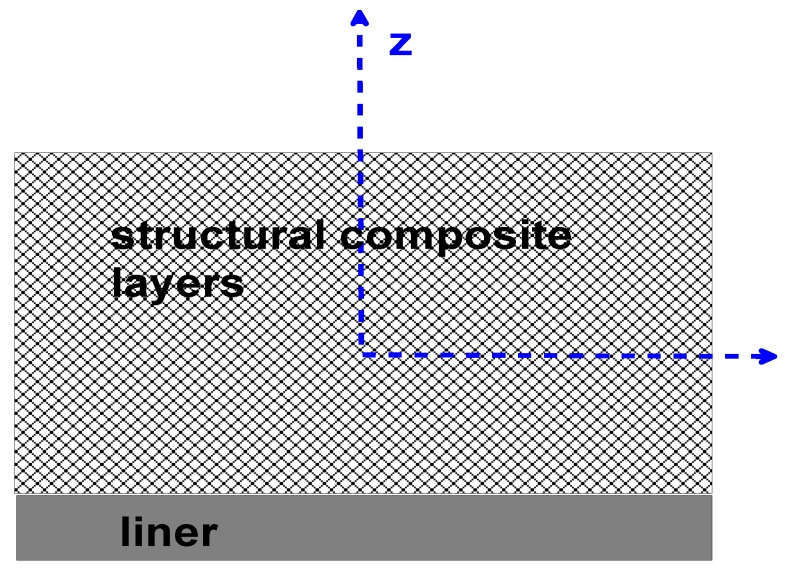
Composition of wall thicknesses of composite pressure vessels.

**Figure 4 materials-13-01792-f004:**
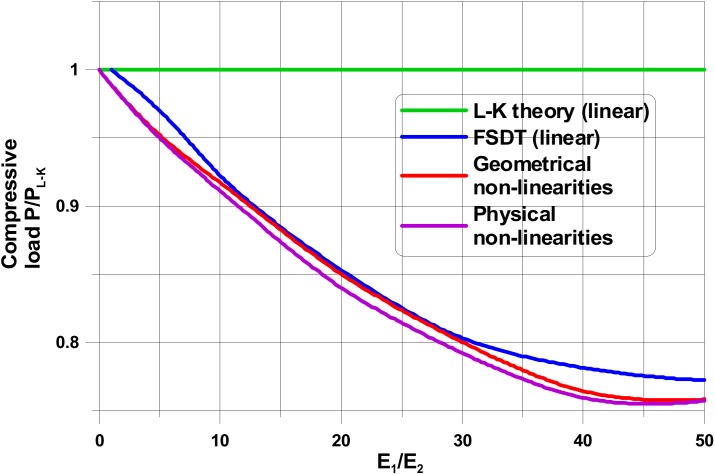
Loads for different variants of theoretical formulations (L/R = 5, R/t = 10, G_12_/E_2_ = 0.5, ν_12_ = 0.25)—the compressed cylindrical shell with a diaphragm.

**Figure 5 materials-13-01792-f005:**
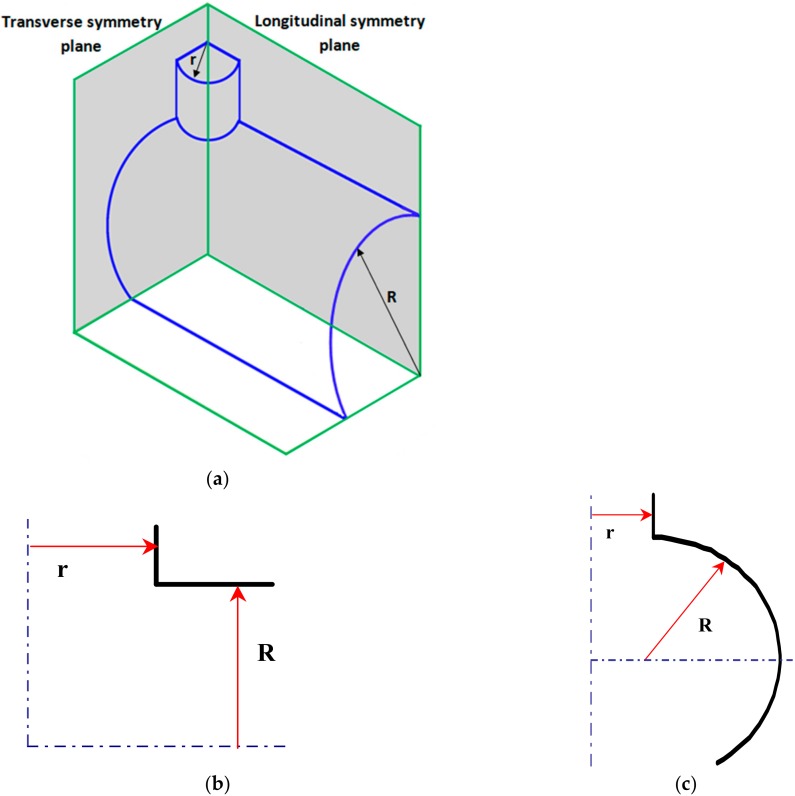
Cross-sections of the junction. (**a**) finite element model of the pressure vessel junction. (**b**) the longitudinal cross-section. (**c**) the transverse cross-section.

**Figure 6 materials-13-01792-f006:**
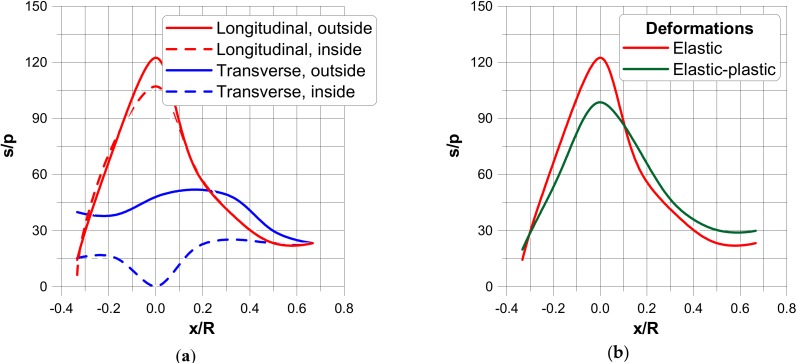
Stress distributions in a direction parallel to the transverse and longitudinal planes. (**a**) elastic deformations. (**b**) comparison of elastic and elastic plastic stress distributions.

**Figure 7 materials-13-01792-f007:**
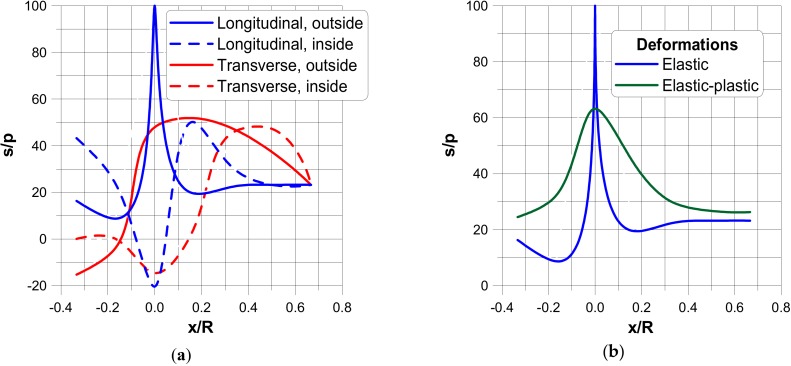
Stress distributions in a direction normal to the transverse and longitudinal planes. (**a**) elastic deformations. (**b**) comparison of elastic and elastic plastic stress distributions.

**Figure 8 materials-13-01792-f008:**
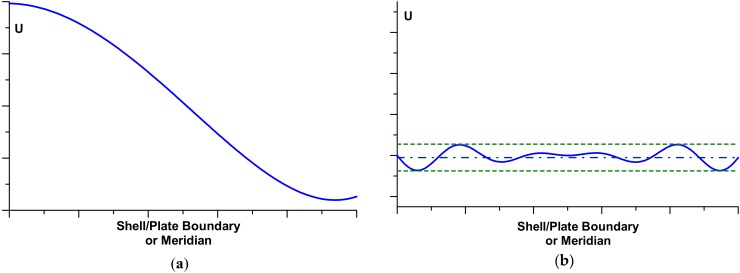
Variations of the objective functional *U* (the strain energy) along the nozzle opening. (**a**) initial. (**b**) final (optimal).

**Figure 9 materials-13-01792-f009:**
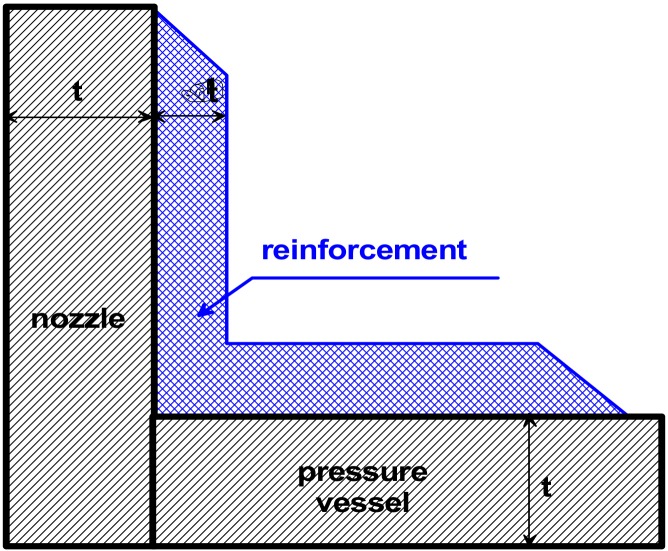
Schematic view of reinforcement—the longitudinal cross-section.

**Figure 10 materials-13-01792-f010:**
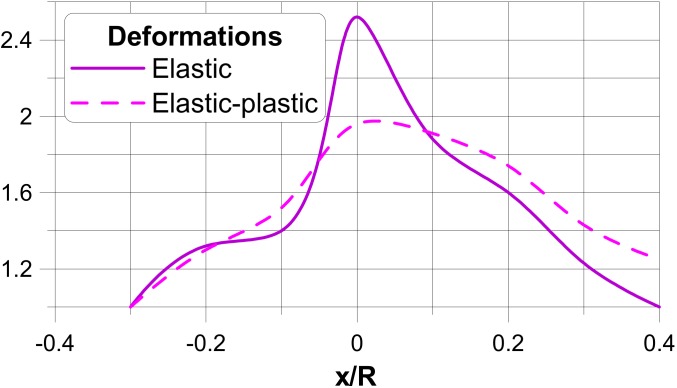
Distributions of the thickness reinforcement in the transverse direction.

**Figure 11 materials-13-01792-f011:**
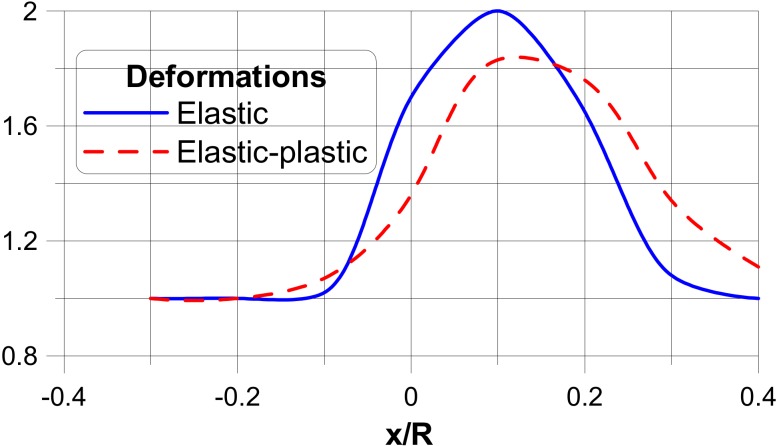
Distributions of the thickness reinforcement in the longitudinal direction.

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
