# Peer review of "Effects of Material Nonlinearities on Design of Composite Constructions—Elasto-Plastic Behaviour"

_materials, 2020, doi:10.3390/ma13071792_

Round 1

Reviewer 1 Report

The manuscript describes the effects of material non-linearities (specifically elasto-plastic behavior) on the design of composite materials. The authors utilize cylindrical shells and junction reinforcements at cones/nozzles to highlight this analysis. 

The author is considered an expert in this field and the manuscript represents a continued contribution to advance the field.  In reading the paper, this reviewer found a few areas confusing in the writing. This may be the result of the authors extensive knowledge, but a few points of added clarification can greatly alleviate these issues. 

I would recommend publishing the paper after these additions. 

  1. There are several places with misspelled words, extraneous hyphens, font mismatch, and single sentence paragraphs. The manuscript should be thoroughly proofread to alleviate these issues. 
  2. The experimental results section is not consistent.
    1. Half of Figure 1 is F vs. d and the second half is stress vs. strain.
    2. The x-axis in Figure 1a and 1b is not readable. 
    3. The composite recipes are not clearly defined from this experimental work. Without extensive reading of the author's prior art, it is difficult to know if these are apples to apples comparisons. For example: Layup recipe (unidirectional, how many layers, [0/90]x or [+-45]). Was the thickness of each composite the same or the number of layers? The authors mention ASTM standards, but there is no reference. I am assuming ASTM 3039. Is the epoxy the same in 1a, 1b, 1c? I see the author is using this to highlight the non-linearity within the tensile curves, but this is not consistent with eqn. 1 or eqn. 7 (stress vs. strain). 
    4. The derivation of eqn. 7 is fairly involved. Given this is a Materials and not composites journal. It would be good to provide a direct reference to this prior work for the reader. 
    5. Figure 2. It is difficult to see the L-K theory because it coincides with the upper graph bound box. I suggest increasing x-axis or changing display of LK in order to remove confusion for reader.
    6. Figure 3 requires some sort of coordinate axis labels. It is best the reader is not left to assume the 1-2-3 axis.
    7. Can the authors clarify how the stress distributions calculated in figures 4 and figure 5? Is this the stress distiribution through the cylindrical shell, from the buckling analysis? It would help to better understand the flow of the work.
    8. The centerline does not appear to go through the center of the composite sheet? (x/R does not range between +0.5 and -0.5 in Figure 5)
    9. Can the author elaborate on this in relation to the coordinate system in Figure 3?
    10. At what strain is the stress calculated for figures 4 and 5?
    11. This is likely my confusion continuing during the reading. In Section 3, the author states the plastic problem initiates almost immediately under loading. If this is so, is it possible to then split the problem into two serial contributions or does the author mean to solve the elastic (linear) + the elastic-plastic (non-linear) problem. I believe it is the latter, but again not clear in the manuscript.
    12. Fig 7 and Fig 8 are not clear. What is t+dt? The increase of thickness in specific locations around the nozzle attachement and the lines represent the stress or optimization value?

      I assume I should be comparing 7 and 8 to 4 and 5 to see accounting for physical non-linearities and adjusting thickness changes the optimal reinforcement distributions.

This work is an important contribution. The points detailed in the conclusions are particularly valuable. My concern is that the typical reader of Materials may be overwhelmed with the current presentation of the work. A few details should alleviate any confusion and lead to an impactful result. 

Reviewer 2 Report

This manuscript describes the effect of material nonliearity on composite structure design. Because the methods and results are clearly described, the referee recommends that this manuscript can be accepted for publication ih the present form.

Author Response

Thank you for your work

Reviewer 3 Report

The manuscript is related to the effects of material nonlinearities on the design of composite structures. Physical and geometrical nonlinearities are considered as well as limit states of composite materials.

Comments for the paper are reported below:

  1. Several typos are present in the manuscript for instance in lines: 14-62-168-170-250-274-275-277-280-283-285
  2. Some figures are not clear: axis in Figs. 1 and 4, lines in Fig. 2 and legend in Fig. 8.
  3. line 107 - No validation for formulas

  4. line 96 - composites with 45° work mainly in torsion/shear (in plane) tension in these configurations is very small. What it is stated is not very clear.

Generally for each result, there are some referring, there is a lot of ambiguity around the novelty of the paper, it is not clear whether the results are from experimental data or numerical one.
Also, the experiment was referred to a reference and so just numerical has been done?. No details on how the modelling has been performed is not mentioned. The geometry of the specimen should be presented in this work, but these information are just randomly reported.

Round 2

Reviewer 3 Report

The author review the paper following the comments raised by reviewers.

The scope of the work seems a sort of review paper or technical paper about a very specific aspect not a proper full article.

I don't see a general state-of-the-art review of the topic because most of the cited contributions come from the main author of the work.

Author Response

I have reduced the number of references from 31 to 25 and changed their numbering.
